# A Clinicopathological Review of 203 Cases of Atypical Polypoid Adenomyoma of the Uterus

**DOI:** 10.3390/jcm12041511

**Published:** 2023-02-14

**Authors:** Yue Sun, Lina Tian, Guoyan Liu

**Affiliations:** 1Key Laboratory of Cancer Prevention and Therapy of Tianjin, Department of Gynecologic Oncology, Tianjin Medical University Cancer Institute and Hospital, National Clinical Research Center for Cancer, Tianjin’s Clinical Research Center for Cancer, Tianjin 300060, China; 2Department of Gynecology and Obstetrics, Tianjin Medical University General Hospital, Tianjin 300052, China; 3Tianjin Key Laboratory of Female Reproductive Health and Eugenics, Tianjin Medical University General Hospital, Tianjin 300052, China

**Keywords:** atypical polypoid adenomyoma, treatment, hysteroscopy, progesterone

## Abstract

Objective: To provide a reference for the diagnosis and treatment of atypical polypoid adenomyoma (APA). Methods: This was a retrospective study of 203 APA patients from 2011 to 2021. The clinicopathological characteristics, treatments, and prognosis were analyzed. Results: The average age at diagnosis of APA patients was 39.30 ± 11.01 years, and premenopausal women accounted for 81.3%. Abnormal uterine bleeding or menorrhagia were the most common clinical manifestations of APA. The uterine fundus (78.3%), followed by the lower segment of the uterus (11.8%), was the most common location of the APA lesions. Abnormal blood vessels were seen on the surface of 28 APA tumors. APA can coexist with atypical endometrial hyperplasia (18.2%) and endometrial cancer (10.8%). Immunohistochemical analysis was performed on 99 samples. In the glandular component, ER (94.8%), PR (94.8%), Ki-67 (51.5%), p53 (45.6%), PTEN (18.8%), and mismatch repair proteins (96.4%) were positively expressed. Stromal immunophenotype expression was exhibited as follows: CD10-(89.5%), p16+(86.9%), h-caldesmon-(66.7%), Desmin+(75%), and Vimentin+(88.9%). Fifty-five APA patients received TCR, and 33 of them received adjuvant therapy after the operation. The postoperative recurrence rate (9.1% vs. 36.4%, *p* < 0.05) and malignant transformation rate (3.0% vs. 18.2%, *p* < 0.05) of the treated group were significantly lower than the untreated group. Conclusions: APA usually occurs in women of childbearing age, and the diagnosis is based on pathological morphology. APA has a low malignant potential, and those who have fertility requirements can undergo conservative TCR treatment, supplemented by progesterone treatment after surgery and close follow-up. Total hysterectomy is the treatment of choice for APA patients with atypical endometrial hyperplasia around the lesion.

## 1. Introduction

Atypical polypoid adenomyoma (APA) is considered a rare intrauterine space-occupying lesion, first described by Mazur in 1981 and defined as a lesion composed of atypical endometrial glands and fibromxyomatous mesenchymal components [1]. APA was previously considered a benign lesion and treated conservatively, but there is increasing evidence that APA has a high rate of recurrence or residual disease and that it precedes the development of cancer [2,3]. About 30.1% of APA patients relapsed after surgery, and 8.8% of patients progressed to endometrial cancer [2]. Given the current research data on APA, the clinical management of APA is complex and should be individualized, especially in patients with fertility needs. There is no current consensus on the standard treatment and follow-up for APA. Total hysterectomy is the main treatment modality for APA patients, but conservative treatment can be performed in young patients who wish to preserve their fertility.

In this study, we retrospectively analyzed clinical data from 203 patients who were diagnosed with APA and treated at our hospital between January 2011 and December 2021. The aim of this study was to investigate the clinicopathological features, clinical management, and follow-up of APA.

## 2. Material and Methods

The Department of Gynecology and Obstetrics of Tianjin Medical University General Hospital treats about 6000 patients with intrauterine space-occupying lesions every year, including about 3600 cases of an endometrial polyp and about 10–25 cases of APA.

The study included 203 APA patients diagnosed in our department between January 2011 and December 2021. All cases were histologically examined by two experienced gynecologic pathologists in our hospital. The diagnostic criteria adopted by APA were based on Longacre et al.’s description, which included a biphasic proliferation of architecturally complex and cytologically atypical endometrioid glands with squamous morules and the presence of a typical myofibromatosis stroma [4]. All patients were followed up regularly until 30 June 2022, and they signed the informed consent at follow-up visits. The study was conducted in accordance with the Declaration of Helsinki, and the protocol was approved by the Institute Research Ethics Committee of Tianjin Medical University General Hospital.

Data were collected in a tabulated manner and analyzed statistically using SPSS 26.0 (IBM, Armonk, NY, USA). to determine the significance values. Differences in proportions for categorical variables were assessed for significance using the χ^2^ test or Fisher exact test. A *p*-value < 0.05 was accepted as statistically significant.

## 3. Results

Patients and clinical characteristics of APA are summarized in Table 1. The average age of the patients was 39.30 ± 11.01 years, ranging from 17 to 64 years. The body mass index (BMI) of these patients was 27.53 ± 3.23 kg/m^2,^ and the obesity rate was 41.9%. Premenopausal women accounted for 81.3%, 46 of them were nullipara (22.7%), and 53 of them were nulligravida (26.1%). There was a past history of hormone therapy in 10 cases (4.9%) and a family history of malignancy in 11 cases (5.4%). Complications consisted of hypertension in 60 subjects and diabetes mellitus in 18 subjects. Four patients had a history of breast cancer (radical surgery and chemotherapy followed by tamoxifen treatment). Symptoms consisted of abnormal uterine bleeding or menorrhagia in 144 cases (70.9%), vaginal discharge in 13 cases (6.4%), and dysmenorrhea in 22 cases (10.8%). The other 29 patients (14.3%) were asymptomatic, but they had abnormal ultrasound imaging findings. The results of clinical examination showed that 56 cases (27.6%) were anemic (hemoglobin level < 100 g/L), but no other changes in biochemical test values or serum tumor marker values were found to contribute to APA diagnosis.

All patients underwent transvaginal or transabdominal sonography, and 65 patients underwent pelvic magnetic resonance imaging (MRI) before treatment. The ultrasound findings included endometrial thickening in 58 cases, hyperechoic mass in 5 cases, hypoechoic mass in 14 cases, and heterogeneous echogenic mass in 73 cases. The boundary between APA and peripheral intima was not clear in 66 cases, and blood flow signals were detected in 64 cases. A representative ultrasound image is shown in Appendix A. However, there are no particular image characteristics to identify APA, and it is difficult to make a differential diagnosis from an endometrial polyp or a submucosal myoma.

Hysteroscopy was used to comprehensively assess endometrial lesions. A representative hysteroscopic image is shown in Appendix A. During hysteroscopy, the lesions presented with a polypoid or submucous myomatoid appearance. There were abnormal vessels on the surface of tumors in 28 cases (13.8%). The site of origin of the tumor was the uterine fundus in 159 cases (78.3%), the lower segment of the uterus in 24 cases (11.8%), and the uterine cervix in 20 cases (0.9%). The tumor diameters ranged from 0.5 cm to 10 cm (average, 2.2 cm). The tumor was single in 179 cases (88.2%) and multiple in 24 cases (11.8%). However, all these features are nonspecific and definitive diagnosis depends on the results of pathological and immunohistological examinations. Histologically, all of these tumors were composed of atypical endometrioid glands and a myofibromatosis stroma (Appendix A). The glandular component has endometrioid glands with an abnormal architecture and varying degrees of cellular atypia. A myofibromatosis stroma is derived from myofibromatosis metaplasia of endometrial stromal cells. In histological examinations, the common pathologies associated with APA were proliferative endometrium (25.1%), atypical endometrial hyperplasia (18.2%), endometrial hyperplasia without atypia (18.2%), endometrial polyp (16.3%), secretory endometrium (11.3%), endometrial carcinoma (10.8%), and submucosal myoma (9.4%) (Figure 1).

Immunohistochemical analysis was performed on 99 samples (Appendix A). Estrogen receptor (ER) and progesterone receptor (PR) stainings were performed on 97 samples, and 92 of the glandular components of APA were positive for ER (94.8%) and PR (94.8%). The expression level of Ki-67, a proliferation marker, was examined in the glandular component of APA. It varied greatly in 99 APA patients who were stained with Ki-67, with a labeling index ranging from 3% to 80%. Of the patients, 51.5% had a Ki-67 proliferation index of <30%, 36.4% of them had a Ki-67 proliferation index of 30%–50%, and 12.1% of them had a Ki-67 proliferation index of >50%. P53 expression was detected in 79 cases, and there were 32 cases of wild-type p53 expression and 47 cases of mutation-type p53 expression (4 cases of diffuse strong positive expression and 43 cases of total negative expression). Mutation-type p53 expression was found in 59.5% of the cases. PTEN expression was detected in only 3 of 16 cases (18.8%). Two cases were stained with CDX2, and 50% was positive. The expression of mismatch repair proteins (MLH1, MSH2, MSH6, and PMS2) was examined in 28 APA samples, and they were omnipresent (96.4%, 27/28). In the APA-positive peri glandular stroma, CD10 staining was negative or weak and patchy (89.5%, 34/38). The expression patterns of p16 in the stroma were positive in most cases (86.9%, 53/61). The h-caldesmon expression was analyzed in three APA cases where a negative staining pattern was identified in two cases. Other stromal immunophenotypic markers were expressed as follows: Desmin (+) (75.0%, 3/4) and Vimentin (+) (88.9%, 16/18).

Histopathological analysis confirmed that all of the patients who underwent surgeries were APA-positive. Fifty-five patients underwent hysteroscopic transcervical resection (TCR) to thoroughly remove the lesions, 33 of whom were treated with adjuvant therapy after the operation (Table 2). Twenty-six patients were treated with medroxyprogesterone acetate (MPA), 160 mg daily for at least 6 months, 2 patients were treated with progesterone, 200 mg daily for at least 3 months, 1 patient was treated with a levonorgestrel-releasing intrauterine system (LNG-IUS), 1 patient was treated with oral Diane-35 for 1 year, and 3 patients were treated with GnRH for 6 months. Of these 33 patients, 3 cases relapsed. One patient relapsed within 1 year after initial TCR (6 months after MPA), and pathological findings after hysteroscopic surgery showed that APA was coexistent with focal canceration. This patient was surgically treated with a laparoscopic hysterectomy, bilateral salpingo-oophorectomy, and pelvic lymphadenectomy, which proved to be endometrial cancer by postoperative pathology. One patient relapsed four years after the initial TCR. The patient underwent a second TCR, followed by oral MPA. No recurrence occurred during follow-up. Another one relapsed seven years after the initial TCR and underwent a hysterectomy because of no fertility requirement. Of the 22 patients who did not receive adjuvant therapy, 8 patients relapsed (Table 2). Four patients relapsed within one year after initial TCR, and pathological findings after hysteroscopic surgery showed that APA was coexistent with focal canceration or endometrial cancer. These patients were surgically treated with a laparoscopic hysterectomy, bilateral salpingo-oophorectomy, and pelvic lymphadenectomy, which were pathologically confirmed to be endometrial cancer. One patient relapsed after the initial TCR for 4 months, then started to take oral MPA after the relapse, and relapsed again after the initial TCR for 9 months and 14 months. One patient relapsed one year after the initial TCR, and two patients relapsed four years after the initial TCR. All three patients received conservative surgery. Of the 55 patients, 13 had a subsequent successful pregnancy. All 148 patients underwent hysterectomy because they had no fertility requirement or were complicated with atypical endometrial hyperplasia, endometrial cancer, hysteromyoma, or adenomyosis.

All patients were followed up for a mean period of 58.9 months (range 7–117 months) by regularly undergoing transvaginal ultrasonography every 3–6 months. All 203 patients were alive without any metastatic diseases and subsequent death.

## 4. Discussion

APA is an uncommon and benign uterine tumor, accounting for about 1% of endometrial polyps [5]. The etiology and pathogenesis of APA remain unclear. Considering the rarity of the disease, no prospective clinical trials for APA are available now. Therefore, there is no consensus on the optimal treatment and follow-up for APA. This study reported the experience of our center, describing the criteria used to diagnose APA and the methods used to provide therapeutic management for this disease.

### 4.1. Clinical Features of APA

Case reviews of APA are shown in Table 3. APA usually occurs in women of reproductive age but may also extend to postmenopausal women (17–81 years) [4,6]. The average age of APA patients in this study was 39.3 years, and 81.3% of them were premenopausal women, which was consistent with previous studies. The data for the relationship between pregnancy and delivery history and infertility in the present study showed nearly a quarter (26.1%) of them were nullipara, and the rate of infertility was 22.7%. In Matsumoto’s review of 29 patients, the majority of patients were nullipara (86.2%), and the infertility rate was 35.5% [7], while in the report by Zhu, 12 (30.8%) of the 30 patients whose history of pregnancy and delivery was known were nulligravida [8]. Many patients with APA are nulligravida and/or nullipara, and whether the presence of APA is a cause or an outcome is unclear but suggests an association with infertility.

In this report, APA patients present with increased menstruation or abnormal uterine bleeding in most cases (70.9%). Other symptoms, such as abnormal vaginal discharge and pelvic pain, are less common. Only 14.3% were asymptomatic and accidentally diagnosed with APA when routine examinations suggested abnormalities. Some 41.9% of the patients are accompanied by obesity. These findings are in accord with other values reported in the literature [2,4,5,9,10] (Table 3).

**Table 3 jcm-12-01511-t003:** Case reviews of APA.

Study		Matsumoto [7] *M/%*	Ma [11] *M/%*	Chiyoda [12] *M/%*	Raffone [5] *M/%*	Zhu [8] *M/%*	Wang [13] *M/%*
Number of cases		29	43	35	237	39	44
Age (years)		38.0	56.0	35	−(17–73)	39.6	48.5
Premenopause		93.1%	48.8%	-	85.5	94.9%	72.7%
Nulligravida		75.9%	7.0%	2.9%	18.6%	30.8%	15.9%
Nullipara		86.2%	11.6%	100%	62.9%	46.2%	-
Symptoms	Abnormal uterine bleeding or menorrhagia	75.0%	83.7%	85.7%	66.5%	87.2%	79.5%
	Vaginal discharge	-	-	-	1.1%	-	4.5%
	Dysmenorrhea	27.6%	-	-	-	-	4.5%
	Asymptomatic	-	7.0%	2.9%	2.2%	5.1%	18.2%
Location	Uterine fundus	58.6%	69.3%	45.7%	55.8%	56.4%	-
	Lower segment	34.5%	20.5%	25.7%	32.7%	38.5%	-
	Uterine cervix	6.9%	10.3%	14.3%	11.5%	5.1%	-
Tumor size (cm)		2.3	2.0	2.2	−(0.1–7)	2.1	2.8
Follow-up time (months)		38.9	26.9	34.0		48.1	62.0
Recurrence rate		23.8%	7%	54.3%	28.9%	5%	0%
Malignant transformation rate		-		-	16.6%		

### 4.2. Assistant Examination of APA

APA patients do not have typical specific clinical manifestations. Auxiliary tools such as gynecological ultrasound and MRI are needed to help diagnose the disease. Ultrasound technology is now routinely used for first-level screening in patients with suspected endometrial diseases [14]. Ultrasound of APA can show heterogeneous endometrial thickening, abnormal intrauterine echoes, and blood flow changes without specificity. Recently, it has been reported that APA can be accurately distinguished from other polypoid tumors of the uterus by using three-dimensional power Doppler hysterosonography (3-DPDS), demonstrating the following characteristics: lobular tumor with inhomogeneous echotexture, echogenicity similar to the myometrium, broad base, acoustic shadow, and linear vessel arrangement [15]. In our study, 65 APA patients performed pelvic MRI. On T2-weighted MRI images, APA could appear as a mixture of hyperintensity and hypointensity, with hyperintensity representing hyperplastic endometrial glands and hypointensity representing smooth muscle-dominated stroma. Further contrast-enhanced study shows irregular enhancement of the tumor [16].

Although all patients underwent transvaginal or transabdominal sonography, and even MRI was applied, it is difficult to make a differential diagnosis of APA prior to performing a surgical procedure. Further hysteroscopy is required to comprehensively assess endometrial lesions. APAs have shapes mostly in the form of a polypoid or submucous myomatoid mass and tend to be yellow-brown or off-white in color. The gross appears as border clear, quality medium-sized tumor with a pedicle, which is often confused with endometrial polyps, cervical polyps, uterine myoma, or polypoid carcinomas [4,17]. According to our patient’s hysteroscopic findings, the lesion usually occurs in the uterine fundus, sometimes in the lower uterine segments or cervical canal, but it has also been reported to involve the oviduct in monkeys [7,8,13]. Lesions are usually single, occasionally multiple. Their sizes range from 0.5 to 10 cm, with an average diameter of 2.2 cm. All of these properties are similar to those reported in the reviews by Matsumoto, Zhu, Wong, and Kihara [7,8,18,19] (Table 3). The blood vessels on the surface of the lesions are abundant and thick. However, it is impossible to accurately distinguish APA from endometrial polyps with hysteroscopy. A hysteroscopic biopsy is advisable. Histological examination of APA pathological diagnosis is currently the diagnostic standard [20,21].

### 4.3. Pathological Features of APA

APA is classified as a mixed epithelial and mesenchymal tumor, according to the fifth edition of the WHO classification of female genital tumors published in 2020 [22]. Histologically, APA is composed of a biphasic proliferation of architecturally complex and cytologically atypical endometrioid glands, squamous morules, and a myofibromatosis stroma. The glandular component has endometrioid glands with an abnormal architecture and varying degrees of cellular atypia [23]. Myofibromatosis stroma is derived from myofibromatosis metaplasia of endometrial stromal cells [24]. Morular metaplasia occurs in more than 90% of APA patients but is not common in other benign lesions, so it can be used as a potent marker for the differential diagnosis. All of our 203 patients had these features conformed with the diagnostic criteria used to identify the presence of APA.

Morular metaplasia may mimic a solid growth pattern [25]. Reliance on these features sometimes makes it extraordinarily difficult for even a professional gynecological pathologist to differentiate between APA and myoinvasive endometrioid cancer. In addition, confusion can arise if the initial diagnosis of APA coexists with atypical endometrial hyperplasia and endometrial cancer. In recent years, more and more studies have explored the immunophenotypic and immunohistochemical patterns of APA in order to discover more targeted differential diagnostic markers (Table 4). Findings from some immunohistochemical studies have shown that ER and PR are widely expressed in the APA glandular elements but almost null in squamous morules. The Ki-67 index is extremely low in squamous morules (<1%), but it is much higher in the glandular epithelium (20–30%). Squamous morules in APA are always associated with higher nuclear β-catenin, CDX2 and CD10 expression compared to their glandular components [21,25]. Moreover, squamous morules of APA typically exhibit diffuse nuclear immunoreactivity with SATB2. The stromal components of APA are strongly positive for a-smooth muscle actin (a-SMA) and SATB2, negative or partially positive for CD10 and negative or focal positive for h-caldesmon [21,25,26]. Currently, CD10, h-caldesmon and p16 are the three main stromal markers recommended for differentiating APA from myoinvasive carcinoma. In myoinvasive carcinoma, CD10 was locally expressed in regions immediately adjacent to glands (“fringe-like staining pattern”) but not in the stroma of APA [21,23,27]. Unfortunately, the absence of detection of CD10 in a fringe-like staining pattern does not rule out myoinvasive carcinoma. H-caldesmon is a late marker of smooth muscle differentiation. H-caldesmon is usually negative or focally positive in the stroma of APA, whereas it is diffusely positive in infiltrated myometrium and may be negative or focally positive in the case of stromal reaction [26,28]. P16, another main stromal marker, is diffusely positive in the stromal component in most APA cases [20]. In contrast, p16 is immunostained negatively in the stroma of myoinvasive endometrioid carcinoma, except for one case of localized p16 staining [20]. Our immunohistochemical results were consistent with the above findings.

### 4.4. Malignant Transformation of APA

Although APA was previously considered a benign lesion, more and more cases suggest that APA has malignant potential. Occasionally, APA has been found to coexist with or precede the development of endometrial cancer, which may be either within the APA or elsewhere in the uterus, similar to atypical endometrial hyperplasia [29,30]. A multicenter study showed that the malignant transformation rate of APA was up to 8.8%, much higher than that of endometrial polyps (0.8%) [2,31]. A systematic review of APA management and follow-up showed that the prevalence of endometrial cancer diagnosed at the same time as APA diagnosed or during follow-up was 16%, of which the rate of malignant transformation to endometrial cancer during follow-up was 14%. In addition, the cumulative risk of a diagnosis of endometrial hyperplasia was 27.27% at 14 years of APA follow-up [10]. In our report, the concurrent atypical endometrial hyperplasia rate was 18.2%, and the concurrent endometrial carcinoma rate was 10.8%. So, how do we identify APAs with malignant potential? Firstly, APA and endometrial cancer share many of the same risk factors, including obesity, diabetes, hypertension and estrogen stimulation [19,32]. The common risk factors result in the occurrence of both diseases. Secondly, several molecular alterations associated with atypical endometrial hyperplasia or endometrial cancer can be used for the malignant potential risk stratification of APA since these two lesions have similar molecular alterations. Some APAs exhibit deletions of mismatch repair proteins and MLH-1 promoter hypermethylation [33]. A comprehensive molecular analysis of APAs has observed a loss of PTEN expression [27,34]. Besides, mutations of the CTNNB1 and KRAS genes are also found in some APA cases [27,35]. Conceptually, APA is best viewed as similar to a localized form of atypical hyperplasia. However, the molecular genetic mechanisms that progress from APA to cancer remain largely unknown and need to be further explored. Histomorphologically, Longacre et al. found the architectural complexity in 25 out of 55 APA cases was indistinguishable from that of well-differentiated endometrial adenocarcinoma, and thus proposed that APAs with significantly complex glands (high architectural index) be designated “APA of low malignant potential” (APA-LMP) to highlight the potential risks related to malignant potential and myometrial invasion [4]. Regrettably, this proposal has not yet been further promoted clinically now. In summary, evaluation of risk factors, molecular or genetic alterations, and morphology may help us identify APA with malignant potential, which guides the choice of clinical treatment.

### 4.5. Treatment of APA

There is no current gold standard for the clinical treatment of patients with APA. APA treatment can be individualized according to age, marital status, fertility requirements, personal circumstances and postoperative pathological diagnosis [23,30,36,37]. Total hysterectomy is recommended for menopausal or perimenopausal APA patients [5,7,30]. However, if the patient is young or wishes to preserve the fertility or the uterus, fertility preservation treatment and a careful postoperative follow-up may be an appropriate treatment.

Dilatation and curettage is a conventional conservative approach to the local resection of lesions with limitations. Firstly, the operator is unable to see the entire uterine cavity. Secondly, it may miss small focal lesions or make the whole lesion unavailable for histological examination [38]. Finally, it is difficult to differentiate APA and myoinvasive well-differentiated endometrial carcinoma in a curettage specimen [17]. In this condition, Vilos described the only reported case of APA treated by operative hysteroscopy in 2003 [39]. Since then, hysteroscopy has been widely used in the conservative treatment of APA. TCR four-step technique described by Di Spiezio Sardo et al. has the lowest disease progression and recurrence rate, including step 1 (complete resection of the APA lesion), step 2 (removal of the endometrium adjacent to the lesion), step 3 (removal of the superficial myometrial tissue at the lesion), and step 4 (multiple-spots biopsy of the endometrium in other parts) [11,38]. Therefore, the four-step diagnosis and treatment method represents a good therapeutic option for APA patients in those women who wish to preserve their fertility. In our study, a total of 55 APA patients with fertility requirements underwent TCR, with a postoperative recurrence rate of 20.0% and a 9.1% probability of progression to endometrial cancer. The data was significantly higher than that reported by Zhu [8], which may be related to the lack of further treatment in about 40% of patients after surgery.

Persistent stimulation with estrogen and a lack of progesterone may be the main pathological mechanisms of APA, and the glandular components of APA are strongly positive for ER and PR, which suggests that hormone therapy may have a protective effect in APA patients [40,41]. Chen found that APA patients with a desire to give birth who received progestin therapy after hysteroscopic resection of the lesion had no recurrence [42]. A recent meta-analysis indicated that medications, especially progestogens, are not the first-line treatment but could prevent the recurrence of APA [10]. The most common drug is MPA (200–600 mg per day) [43]. Dihydrogestone, megestrol and an LNG-IUS could also be used as a conservative treatment for APA patients [44]. In this study, only 60% of 55 APA patients with fertility requirements received progestin for at least six months after TCR. The recurrence rate and the rate of progression to endometrial cancer were significantly lower in these patients than in the untreated group after surgery, providing evidence to support the use of progestogen in APA therapy. However, postoperative adjuvant progestin therapy for APA patients remains controversial, and recent studies have shown that adjuvant drug therapy has no clear inhibitory effect on APA recurrence. Mikos’s systematic review analyzed the therapeutic effects of MPA, dydrogesterone, and megestrol in APA patients, but their impacts on APA recurrence and cure rates were not satisfactory [30]. Perhaps, APA patients complicated with endometrial tumors or endometrial hyperplasia would benefit more from progestin therapy. Additional studies are warranted in the future to assess whether the alterations of certain molecular markers can predict APA responsiveness to progestin.

### 4.6. Prognosis and Follow-Up of APA

Patients receiving conservative therapy have a high rate of relapse and endometrial cancer. Once the patient is diagnosed with APA, a long-term follow-up is required. Research has indicated that the recurrence rate of APA patients ranges from 28.9% to 44% [5,10,30], and the rate of progression to endometrial carcinoma is 8.8% to 16.6% after conservative treatment [2,5,30]. Compared with dilatation and curettage, hysteroscopic surgery has lower recurrence and malignancy rates. In terms of reproductive outcomes, existing studies suggest that patients undergoing conservative hysteroscopy, polypectomy or repeat curettage have subsequent successful spontaneous pregnancy [19,41,45,46,47]. Mikos’s review showed a pregnancy rate of 60.0% and a live birth rate of 56.4% among 55 patients wishing to give birth [28]. In Raffone’s report, 25.3% of all patients receiving fertility-sparing treatment achieved pregnancy [5]. A good prognosis was realized, and no deaths were observed in all 203 patients. In terms of reproductive outcomes, 23.6% of APA patients who received conservative treatment achieved pregnancy. A matter of concern is whether APA recurrence and progression rates may be affected by pregnancy. Only one recurrence and one persistent disease were reported in patients with successful pregnancies [4,42]. These findings appear to be consistent with other studies advocating a protective effect of pregnancy on APA [43].

There is also considerable uncertainty in the follow-up of APA patients, with follow-up patterns and durations varying across studies. In three studies by Nomura et al. and Chen et al., curettage or hysteroscopic biopsy specimens were obtained only in the setting of suspicious findings on ultrasonography [42,43,48], while in others, histology was performed at each follow-up visit [7,11,12,49]. Given the risk of APA recurrence, progression, and coexisting endometrial atypical hyperplasia and carcinoma, close follow-up based on histology is desirable. Nomura et al. believed that two to five years after surgery was the peak time for APA recurrence or canceration. Therefore, a close follow-up for five years is recommended after surgery [43]. The follow-up pattern described by Nomura et al. consists of curettage or hysteroscopic biopsy combined with transvaginal ultrasonography every three months for the first two years, transvaginal ultrasonography every four to six months for another three years and transvaginal ultrasonography once a year thereafter [43,48]. Moreover, if the patient develops abnormal uterine bleeding, intrauterine space-occupying lesions and other symptoms during the follow-up period, hysteroscopy and endometrial biopsy should be performed at the same time.

## 5. Conclusions

APA is a rare uterine tumor that occurs in women of childbearing age. Abnormal uterine bleeding, anemia, and infertility are the main clinical symptoms of APA. Preoperative examinations, including B-ultrasound, MRI and even PET-CT, can only be used for preliminary evaluation of uterine conditions, and pathological examination under hysteroscopy is advisable. Detection of immunohistochemical markers and the appearance of squamous metaplasia can aid in the diagnosis and differential diagnosis of APA. Although histologically benign, it has a tendency to recur and become malignant. Thus, simple hysterectomy is the treatment of choice in postmenopausal women, while in patients who desire to become pregnant or preserve the uterus, hysteroscopy with complete excision of the lesions should be the preferred treatment. The patients should be treated individually, followed up closely, and undergo regular hysteroscopy and endometrial biopsy.

## Figures and Tables

**Figure 1 jcm-12-01511-f001:**
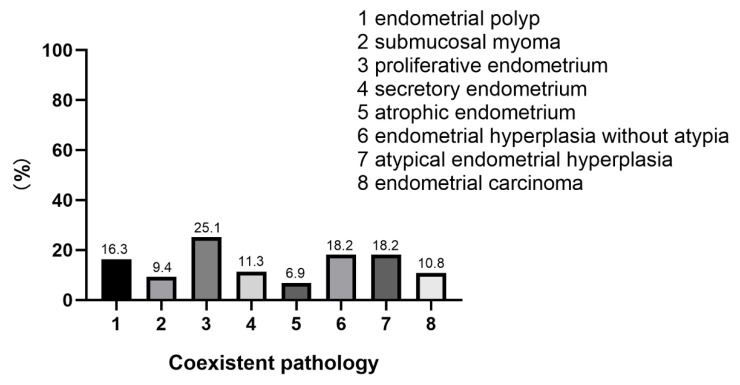
The coexistent pathologies associated with APA.

**Table 1 jcm-12-01511-t001:** Clinical characteristics of APA in our study.

Clinical Characteristics		Our Study *M/x* ± *s/%(N)*
Age (years)		39.3 (17–64)
BMI (kg/m^2^)		27.53 ± 3.23
Premenopause		81.3% (165/203)
Nulligravida		22.7% (46/203)
Nullipara		26.1% (53/203)
History of hormone therapy		4.9% (10/203)
Family history of malignancy		5.4% (11/203)
Complications	Hypertension	29.5% (60/203)
	Diabetes	8.9% (18/203)
	Breast cancer	1.8% (4/203)
Symptoms	Abnormal uterine bleeding or menorrhagia	70.9% (144/203)
	Vaginal discharge	6.4% (13/203)
	Dysmenorrhea	10.8% (22/203)
	Asymptomatic	14.3% (29/203)
Location	Uterine fundus	78.3% (159/203)
	Lower segment	11.8% (24/203)
	Uterine cervix	9.9% (20/203)
Tumor size (cm)		2.2 (0.5–10)
Survival rate		100.0% (203/203)
Follow-up time (months)		58.9 (2–137)
Recurrence rate		20.0% (11/55)
Malignant transformation rate		9.1% (5/55)

**Table 2 jcm-12-01511-t002:** Effect of adjuvant therapy after conservative surgery in APA.

Prognosis	APA		*p*
	Treated Group (n = 33)	Untreated Group (n = 22)	
	*N* (%)	*N* (%)	
Relapse	3 (9.1%)	8 (36.4%)	0.013
Malignant transformation	1 (3.0%)	4 (18.2%)	0.046
Pregnancy	8 (24.2%)	5 (22.7%)	0.897

**Table 4 jcm-12-01511-t004:** Immunohistochemical features of APA.

APA		Immunohistochemical Markers
Glands		ER (+), PR (+), Ki-67 index 20–30% (+) CD10 (−)
	squamous morules	ER/PR/Ki-67 index: <1%(+) β-catenin (+), CDX2 (+), SATB2 (+), CD10 (+)
Stroma		SMA (+), h-caldesmon (−) or focal (+), CD10 (−) or partially (+), SATB2 (+), p16 (+), Desmin (+)

## Data Availability

Data is contained within the article or Appendix A.

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
