# Peer review of "A Clinicopathological Review of 203 Cases of Atypical Polypoid Adenomyoma of the Uterus"

_jcm, 2023, doi:10.3390/jcm12041511_

Round 1

Reviewer 1 Report

Thank you for this comprehensive review and analysis of your collective of patients with atypical polypoid adenomyoma. To increase its value for the inclined reader I recommend significant shortening of the article by laying focus to the key points, possibly transfering some information to additional tables or figures and omitting them in the text aside from a reference to the table / figure of course and introducting of various subheadings at the appropriate locations in the text to further structure the article. All tables should be revised as they are not organized enough at the moment. 

Reviewer 2 Report

Congratulate for exelent approach in description a rare gynecological pathology issue.

My suggestion as You approach to explain all details in APA to put hystoroscopy image, UZ image, hystological image to get full picture to who will read this article.Of course if it is allowed by the Journal.

Also I suggest to use new WHO classifiction benigne hyperplasia and atypical endometrial hyperplasia dual classification.

Best regards

Reviewer 3 Report

This is a nice study of a tumor with only few available information in the litterature.

Comments:

1/ This is a retrospective study. How the informed consent before inclusion was obtained?

2/ The number of cases is enormous for this lesion even for a 10-years period. It is not by chance that there are no large series in the litterature. The authors should explain this by using epidemiological data of their region.

3/ This number corresponded at what % of all curetages, of malignant cases, of atypia cases etc?

4/ Histological analyses were performed in the curetage specimens, or the hysterectomies also?

5/ The DD criteria used to differentiate from atypical hyperplasia and cancer should be exhaustively explained.

6/ Why MMR proteins are examined in the stroma?????

7/ What does p53 expression mean? Either there is a mutated profile or not.

8/ The authors should notice in the abstract and text that only a part of lesions were IHC examined.

9/ The authors should provide a comprehensive Table with the most imporant series in the topic and their findings.

Round 2

Reviewer 3 Report

The authors respond to some of the comments, but their description is still inaccurrate in several points since they do not incorporate the changes in the text (!) :

1/ The abstract should explicitely describe the number of cases where IHC was available.

2/ All the information in the responses to comments, for ex the number of cases treated in their hospital, when the patients were informed and signed, should ALSO appear IN the text. 

3/ Similarly, the authors do not explain in th MM, that this was just a relecture of the files, no relecture of slides etc was perfomed. These details should be noted. Similarly, they cannot declare in the results "we then performed IHC in some samples", since this was obviously already done during routine diagnosis and just reused here as an information.

4/ P53 results are still inaccurate. The authors just respond which is the definition of p53 mutated profile (and even this, it is not correct, not only strong diffuse expression is mutated!) by they do not explain in the text, how many were mutated, just p53 detected in 36 cases. If they don't know, they should say it so.

5/ The ki67 results are not clear either: stroma or glnads stianing? And 80% as result is enormous for this entity.
